# TiO₂/ZnO Nanofibers Prepared by Electrospinning and Their Photocatalytic Degradation of Methylene Blue Compared with TiO₂ Nanofibers

**Chang-Gyu Lee [1], Kyeong-Han Na [1], Wan-Tae Kim [1]** 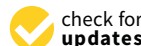**, Dong-Cheol Park [1,2], Wan-Hee Yang [2] and Won-Youl Choi [1,3],\***

[1] Department of Advanced Materials Engineering, Gangneung-Wonju National University, Gangneung, Gangwon 25457, Korea

[2] WITH M-TECH Co., Ltd., Suwon, Gyeonggi 16229, Korea

[3] Research Institute for Dental Engineering, Gangneung-Wonju National University, Gangneung, Gangwon 25457, Korea

\* Correspondence: cwy@gwnu.ac.kr

**Abstract:** $TiO_2$ nanofibers have high chemical stability and high strength and are applied to many fields such as air pollution sensors and air pollutant removal filters. ZnO nanofibers also have very high absorptivity in that air and are used as germicides and ceramic brighteners. $TiO_2$/ZnO nanofibers, which have a composite form of $TiO_2$ and ZnO, were fabricated and show higher photocatalytic properties than existing $TiO_2$. The precursor, including zinc nitrate hexahydrate, polyvinyl acetate, and titanium isopropoxide, was used as a spinning solution for $TiO_2$/ZnO nanofibers. Electrospun $TiO_2$/ZnO nanofibers were calcined at 600 °C and analyzed by field emission scanning electron microscope (FE-SEM) and X-ray diffraction (XRD). The average diameter of $TiO_2$/ZnO nanofibers was controlled in the range of 189 nm to 1025 nm. XRD pattern in $TiO_2$/ZnO nanofibers have a $TiO_2$ anatase, ZnO, $Ti_2O_3$, and $ZnTiO_3$ structure. $TiO_2$/ZnO nanofibers with a diameter of 400 nm have the best photocatalytic performance in the methylene blue degradation experiments and an absorbance decrease of 96.4% was observed after ultraviolet (UV) irradiation of 12 h.

**Keywords:** $TiO_2$ nanofibers; $TiO_2$/ZnO nanofibers; electrospinning; photocatalyst

## 1. Introduction

There is a growing concern about air pollution due to internal combustion engines installed in vehicles and the large amount of exhaust gas generated in various industrial fields. Typical air pollutants include volatile organic compounds which are known to be very harmful to the human body. The World Health Organization (WHO) has announced that the incidence of various respiratory diseases can be lowered by reducing air pollution. Air pollution will become more and more serious as the scale of various industries increases. Accordingly, various solutions for reducing the concentrations of pollutants in the air such as microorganisms, photocatalysts, and activated carbon have been proposed. Among them, photocatalyst technology has attracted attention because it can continuously decompose pollutants by using renewable energy [1–3]. A photocatalytic method has been studied as a method of removing pollutants. A photocatalyst means that the oxidation-reduction reaction on the surface of a substance occurs by absorbing light. It is applied to various tasks such as decomposition of harmful substances by the oxidation-reduction reaction that occurs on the surface, gas detection, ultraviolet blocking, antibacterial function, and midnight function. The photocatalytic reaction does not require other energy because the reaction is also caused by solar energy and light. Since $CO_2$ and $H_2O$ generated by these photocatalytic reactions do not pollute the environment,

they are environmentally friendly and economical because they can be used semipermanently [4–7]. Many $TiO_2$ nanomaterials and ZnO nanomaterials are used as these photocatalyst materials [8–15]. This is because $TiO_2$ nanomaterials are very stable physically and chemically and have excellent heat resistance and bio-friendly properties [16–19]. ZnO nanomaterials have a high degree of adsorption and are used as ceramic brighteners, germicides, and the like [20–23].

Nanomaterials such as nanofibers, nanotubes, and nanoparticles have various physical and chemical properties because they have various structures and pore distribution [24–26]. Among such nanomaterials, one-dimensional nanomaterials have different properties from bulk materials and have been studied in various fields for the application of chemical sensors, photovoltaic cells, optical filters, and improvement of catalytic activity [27–33]. Nanofibers have excellent photocatalytic and electrical properties because of their large specific surface area and one-dimensional structure [34–37]. The electrospinning process is a practical technique with a low cost and high efficiency, and many studies have reported producing these various nanofibers [38–43]. In the electrospinning process, the precursor solution flows through at a constant rate through a pump in such a way as to create a continuous nanofiber; then electrodes are connected to the inflowing electrospinning solution and other electrodes are connected to the appliance plate. At this time, if a high voltage is applied, it is emitted in a conical shape by surface tension at the electrospinning solution end. The charge is subsequently stored in the electrospinning solution, and the mutual repulsion causes the cone to be radiatively stretched to jet when the surface tension of the electrospinning solution is exceeded. In the radiation-stretched electrospinning solution, volatilization of the solvent occurs before it collects in the plate, and it is possible to obtain disorderly arranged nanofibers in the plate [44,45].

In this study, the electrospinning process was used to fabricate $TiO_2$/ZnO nanofibers for photocatalyst materials and process variables were controlled to obtain a stable and optimized one-dimensional structure. The microstructure and phase of the electrospun $TiO_2$/ZnO nanofibers were analyzed through field emission scanning electron microscopy (FE-SEM) and X-ray diffraction (XRD), respectively. The photocatalytic efficiencies of $TiO_2$/ZnO nanofibers and $TiO_2$ nanofibers were observed by ultraviolet–visible spectroscopy (UV–Vis) using the photocatalytic decomposition of methylene blue.

## 2. Materials and Methods

### 2.1. Electrospinning Process

The solution, *N,N*-Dimethylformamide (DMF, extra pure, DAEJUNG Chemicals and Metals Co., Ltd., Siheung-Si, Korea), was used as a solvent for dissolving a polyvinyl acetate (PVAc, Mw~500,000 by GPC, Powder, Sigma-Aldrich Co., Ltd., St. Louis, MO, USA) and 5 wt % PVAc was dissolved. DMF solution of 5g with PVAc was stirred for 4 h using a stirrer. After dissolution was complete, 6 g titanium (IV) isopropoxide (TTIP, JUNSEI Co., Ltd., Tokyo, Japan) and 21–35 wt % acetic acid (extra pure, DAEJUNG Chemicals and Metals Co. Ltd., Siheung-Si, Korea) were added. The solution became clear and 0.1 g zinc powder (Powder, Sigma-Aldrich Co., Ltd., St. Louis, MO, USA) was added and stirred for 2 h.

Figure 1 is a diagram schematically showing the electrospinning process. The distance between the needle and the plate was 15 cm and a 10-mL syringe was used. To obtain the various microstructures of $TiO_2$/ZnO nanofibers, experimental variables such as voltage, inflow rate, and amount of acetic acid were controlled. The applied voltages of 12 kV, 15 kV, and 18 kV, the pump inflow rates of 1 mL/h, 0.8 mL/h, and 0.6 mL/h, and 21 wt %, 28 wt %, and 35 wt % of acetic acid were used, respectively. The electrospinning process variables for $TiO_2$/ZnO nanofibers are summarized in Table 1.

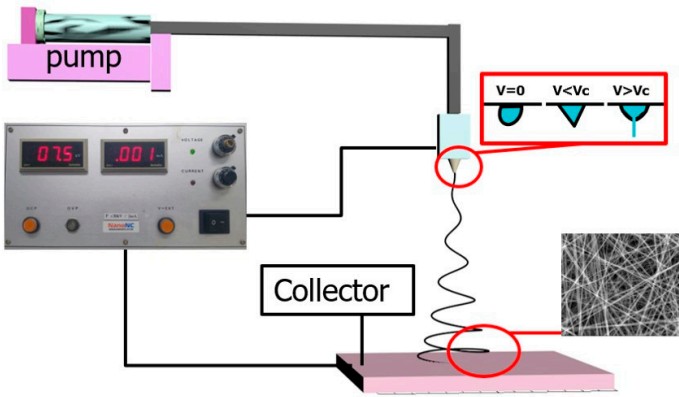

**Figure 1.** Schematic diagram of electrospinning process.

**Table 1.** Process variables of electrospinning for $TiO_2$/ZnO nanofibers.

| Variable | Voltage (kV) | Flow Rate (mL/h) | Acetic Acid (wt %) |
|---|---|---|---|
| | 12 | 1 | 21 |
| Voltage | 15 | 1 | 21 |
| | 18 | 1 | 21 |
| | 12 | 1 | 21 |
| Flow rate | 12 | 0.8 | 21 |
| | 12 | 0.6 | 21 |
| | 12 | 1 | 21 |
| Acetic acid | 12 | 1 | 28 |
| | 12 | 1 | 35 |

Electrospun $TiO_2$/ZnO nanofibers were dried at 80 °C for 24 h in an oven. The dried $TiO_2$/ZnO nanofibers was heat-treated at 600 °C for 1 h in a furnace. The microstructure of $TiO_2$/ZnO nanofibers was observed using FE-SEM (SU-70, Hitachi Co., Tokyo, Japan) and the diameters of $TiO_2$/ZnO nanofibers were measured. To confirm the metal oxide phase of $TiO_2$/ZnO nanofibers, XRD (AXS-D8, Bruker Co., Madison, WI, USA) patterns were also observed from 20° to 80° at a rate of 0.03° per step using a Cu Kα target.

## 2.2. Evaluation of Photocatalytic Efficiency

The photocatalytic decomposition reaction was carried out using an ultraviolet lamp (JINLED Co., Ltd.) of 10 W at room temperature as an irradiation light source. The irradiation distance between the lamp and the sample was fixed to 10 cm. A nanofiber of 0.5 g was added to the methylene blue aqueous solution, and the mixture was stirred to irradiate the light source. To compare $TiO_2$/ZnO nanofibers with $TiO_2$ nanofibers, $TiO_2$ nanofibers were fabricated by the electrospinning process. The electrospinning process conditions for the nanofibers are shown in Table 2.

**Table 2.** Electrospinning process conditions of nanofibers used in the decomposition of methylene blue for photocatalytic performance evaluation.

| | Voltage (kV) | Flow Rate (mL/h) | Acetic Acid (wt %) |
|---|---|---|---|
| $TiO_2$ nanofibers | 15 | 1 | 0 |
| $TiO_2$/ZnO-1 nanofibers | 15 | 3 | 28 |
| $TiO_2$/ZnO-2 nanofibers | 15 | 2 | 28 |

To fabricate $TiO_2$ nanofibers, polyvinylpyrrolidone (PVP, MW 1,300,000 Powder, ALFA AESAR Co., Ltd., Tewksbury, MA, USA) and ethyl alcohol (EtOH, 99.5%, SAMCHUN Chemical Co., Ltd.,

Seoul, Korea) were used. TTIP and acetylacetone (ACAC, JUNSEI Chemical Co., Ltd., Tokyo, Japan) were also added into the PVP-based solution and the mixed solution was stirred for 2 h. The prepared $TiO_2$ electrospinning solution was mounted on the pump and the electrospinning was conducted at the pump inlet rate of 1 mL/h and voltage of 15 kV. As-spun $TiO_2$ nanofibers were heat-treated at 450 °C for 3 h and $TiO_2$ nanofibers with an anatase phase were prepared to compare with $TiO_2$/ZnO nanofibers. The molar ratio of $TiO_2$ to ZnO was equal to 17:1 in $TiO_2$/ZnO-1 and $TiO_2$/ZnO-2 nanofibers. To obtain a different diameter, the flow rate was controlled. $TiO_2$/ZnO-1 and $TiO_2$/ZnO-2 nanofibers were fabricated at a flow rate of 3 mL/h and 2 mL/h, respectively. The prepared $TiO_2$/ZnO nanofibers were dried at 80 °C for 24 h. The dried specimen was placed in a heating furnace and heat-treated at 600 °C for 1 h. To evaluate the photocatalytic activity, UV–Vis spectrums with the $TiO_2$ and $TiO_2$/ZnO nanofibers were observed using the photocatalytic decomposition of methylene blue.

## 3. Results

### 3.1. $TiO_2$/ZnO Nanofibers

Figure 2 shows the FE-SEM images of $TiO_2$/ZnO nanofibers fabricated by the electrospinning process at various applied voltages of 12 kV, 15 kV, and 18 kV. $TiO_2$/ZnO nanofibers have a typical one-dimensional and clean microstructure without droplets. Increasing the voltage during the electrospinning process increased the charge density of the solution and nanofibers with a small diameter were obtained. When voltage exceeding the limit was applied, nanofibers were not formed and sprayed. If a lower voltage is used then surface tension is applied, and nanofibers in the form of nanofiber intermediate beads can be obtained instead of nanofibers of a constant diameter.

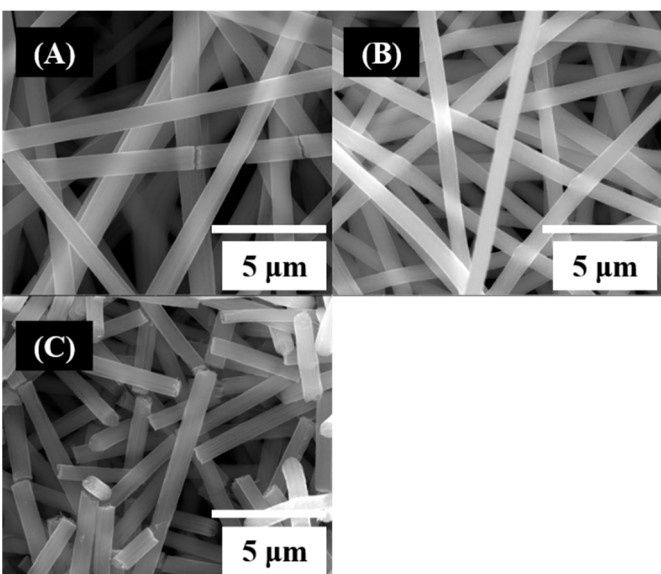

**Figure 2.** Field emission scanning electron microscope (FE-SEM) images of electrospinning $TiO_2$/ZnO nanofibers: (**A**) 12 kV, (**B**) 15 kV, and (**C**) 18 kV.

The average diameters of $TiO_2$/ZnO nanofibers with applied voltage was 1025 nm, 735 nm, and 694 nm, respectively. It reveals that the higher the voltage, the smaller the diameter. The diameters of $TiO_2$/ZnO nanofibers with applied voltage are plotted in Figure 3.

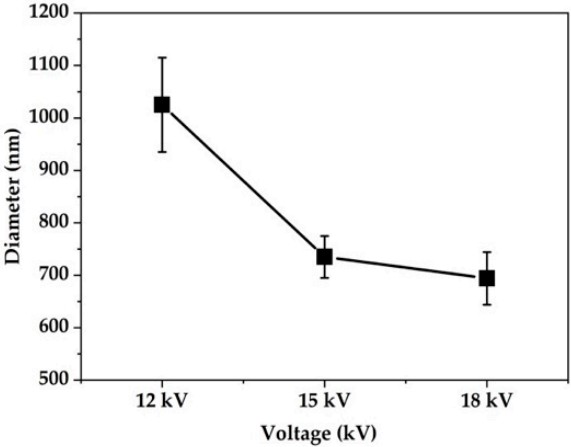

**Figure 3.** Average diameters of TiO$_2$/ZnO nanofibers with voltage.

Figure 4 shows FE-SEM images of TiO$_2$/ZnO nanofibers fabricated by the electrospinning process at various inflow rates of 1.0 mL/h, 0.8 mL/h, and 0.6 mL/h. A one-dimensional nanofiber was observed in all TiO$_2$/ZnO nanofibers. When the flow rate of the electrospinning solution is fast, the size of the droplet that forms on the tip of the needle becomes large. Even when the solution is reached at the collector, the solvent in the solution may not be completely evaporated, and a nanofiber with a bead-shape or large diameter can be obtained.

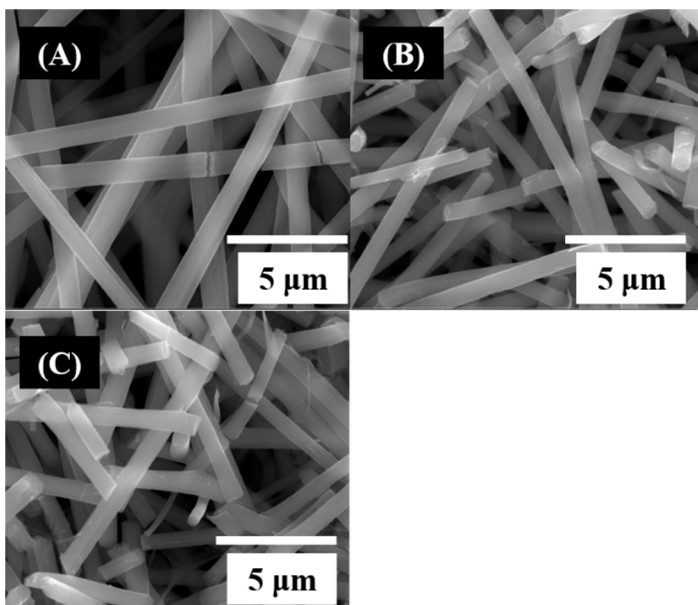

**Figure 4.** FE-SEM images of electrospinning TiO$_2$/ZnO nanofibers: (**A**) 1 mL/h, (**B**) 0.8 mL/h, and (**C**) 0.6 mL/h.

The average diameters of TiO$_2$/ZnO nanofibers with an inflow rate were 1025 nm, 771 nm, and 728 nm, respectively. The average diameter of TiO$_2$/ZnO nanofibers increased with inflow rate. The diameters of TiO$_2$/ZnO nanofibers with an inflow rate are plotted in Figure 5.

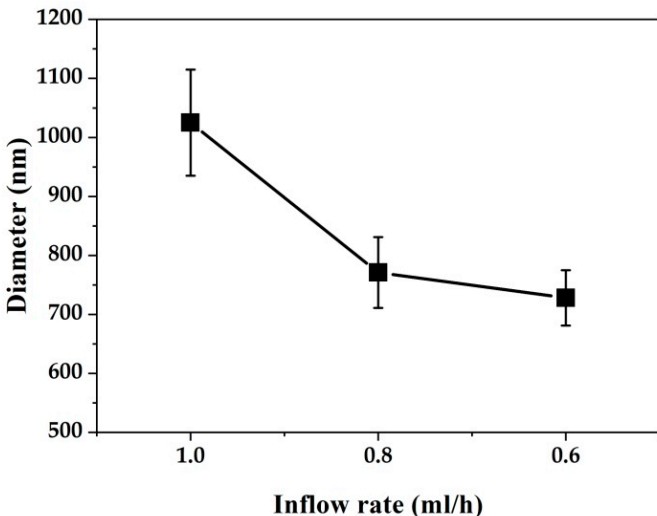

**Figure 5.** Average diameters of TiO$_2$/ZnO nanofibers with inflow rate.

Figure 6 shows FE-SEM images of TiO$_2$/ZnO nanofibers fabricated by the electrospinning process at various amounts of acetic acid—21 wt %, 28 wt %, and 35 wt %—in the precursor solution. All TiO$_2$/ZnO nanofibers have nanofiber structural networks. As the amount of acetic acid in the electrospinning solution increases, the viscosity decreases, and the diameter of the nanofibers dramatically decreases. This reveals that the viscosity is the most important parameter for controlling the diameter in the electrospinning process.

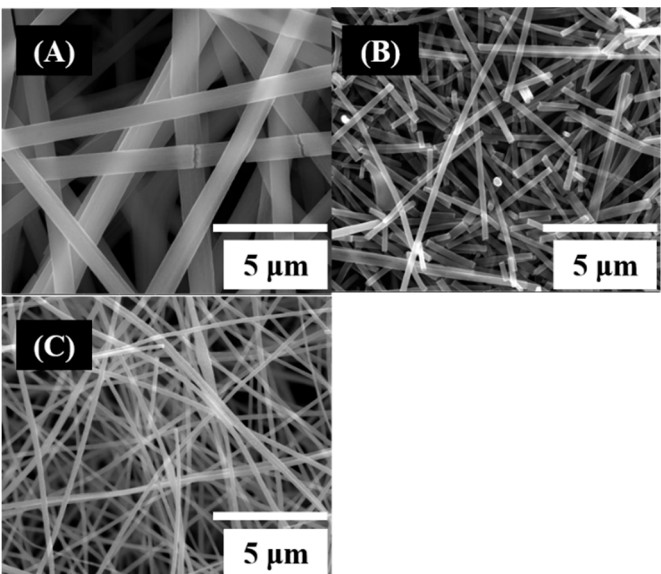

**Figure 6.** FE-SEM images of electrospinning TiO$_2$/ZnO nanofibers: (**A**) 21 wt %, (**B**) 28 wt %, and (**C**) 35 wt %.

The average diameters of TiO$_2$/ZnO nanofibers with an inflow rate were 1025 nm, 233 nm, and 189 nm, respectively and the average diameter dramatically decreased with the amount of acetic acid. The diameters of TiO$_2$/ZnO nanofibers with different amounts of acetic acid are plotted in Figure 7.

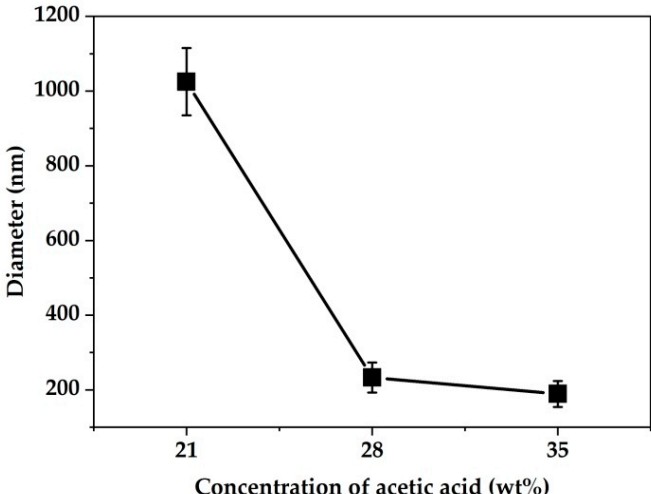

**Figure 7.** Average diameters of TiO$_2$/ZnO nanofibers according to the amount of acetic acid.

To observe the crystal phase of TiO$_2$/ZnO nanofibers, XRD analysis was conducted. Figure 8 shows the XRD pattern of TiO$_2$/ZnO nanofibers. The peaks of TiO$_2$ anatase, ZnO, Ti$_2$O$_3$, and ZnTiO$_3$ were identified and the peak of ZnTiO$_3$ was obtained by the reaction of TiO$_2$ and ZnO.

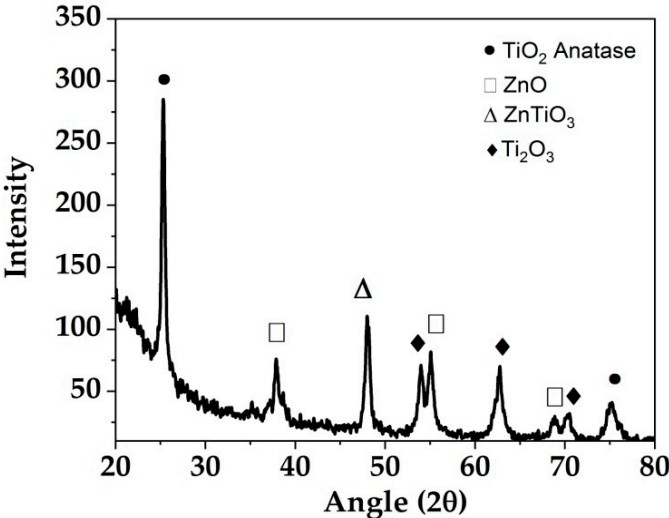

**Figure 8.** X-ray diffraction (XRD) pattern of TiO$_2$/ZnO nanofibers.

## 3.2. Photocatalytic Performance Evaluation

The photocatalytic reaction of TiO$_2$/ZnO is schematically shown in Figure 9. The electrons in the valence band can move to the conduction band and generate holes in the valence band by ultraviolet (UV) rays in sunlight. In heterojunction of ZnO with an energy band gap of 3.2 eV and TiO$_2$ with an energy band gap of 3.4 eV, the electrons in the conduction band of ZnO move to the conduction band of TiO$_2$; then the holes move from the valence band of TiO$_2$ to the valence band of ZnO. Electrons transferred to the TiO$_2$ conduction band react with O$_2$ to form a superoxide anion (O$_2^-$). The holes transferred to the ZnO valence band react to form a strong hydroxyl radical (OH$^\bullet$). Such reactions continuously occur under UV irradiation, and high active radicals such as O$_2^-$ and OH$^\bullet$ can promote the photodegradation of organic pollutants [46–49]. Photodegradation characteristics using methylene blue were investigated for photocatalytic performance evaluation of TiO$_2$/ZnO nanofibers. TiO$_2$ nanofibers were used as a reference material to compare with TiO$_2$/ZnO nanofibers. All nanofibers were fabricated by the electrospinning process.

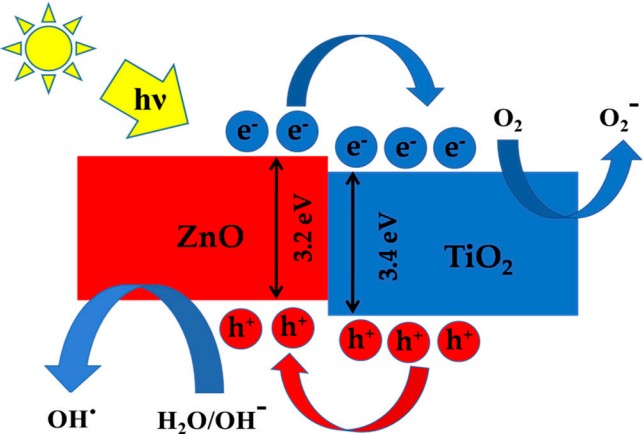

**Figure 9.** Illustration of photocatalytic mechanism in TiO$_2$/ZnO heterojunction [46].

Figure 10 shows the FE-SEM images of TiO$_2$ nanofibers and TiO$_2$/ZnO-1 and TiO$_2$/ZnO-2 nanofibers used in methylene blue photolysis experiments for photocatalytic performance evaluation. The microstructure in all nanofibers shows a typical one-dimensional nanofiber structure and their shapes are similar to each other. The diameters of TiO$_2$, TiO$_2$/ZnO-1, and TiO$_2$/ZnO-2 nanofibers are 400 nm, 600 nm, and 400 nm, respectively. The diameters of TiO$_2$ nanofibers and TiO$_2$/ZnO-1 and TiO$_2$/ZnO-2 nanofibers are summarized in Table 3.

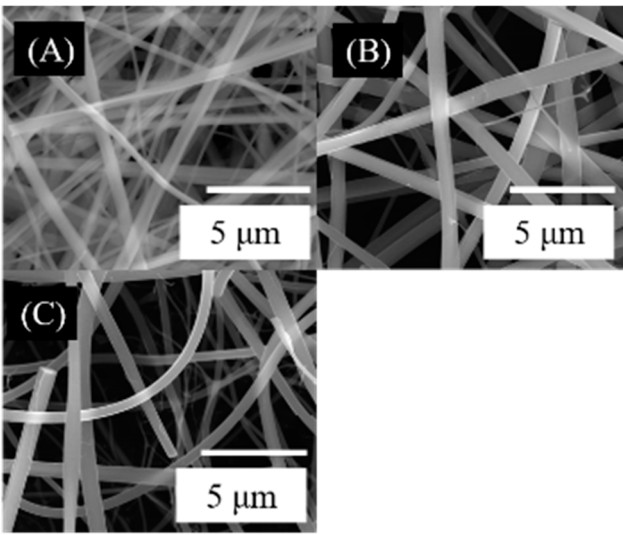

**Figure 10.** FE-SEM images of electrospinning process conditions of nanofibers used in the decomposition of methylene blue for photocatalytic performance evaluation: (**A**) TiO$_2$ nanofibers, (**B**) TiO$_2$/ZnO-1 nanofibers, (**C**) TiO$_2$/ZnO-2 nanofibers.

**Table 3.** The nanofibers used in the methylene blue decomposition experiments for photocatalytic performance evaluation.

| Nanofibers | Diameter (nm) |
|---|---|
| TiO$_2$ | 400 ± 32 |
| TiO$_2$/ZnO-1 | 600 ± 27 |
| TiO$_2$/ZnO-2 | 400 ± 13 |

Figure 11 shows the absorption spectra to evaluate the photocatalytic performance of TiO$_2$/ZnO nanofibers observed by UV–Vis analysis. Figure 11a–d are the spectra of methylene blue without nanofibers, with TiO$_2$ nanofibers, with TiO$_2$/ZnO-1 nanofibers, and with TiO$_2$/ZnO-2 nanofibers,

respectively. A total of 0.5 g of nanofiber was added in a 200 mL methylene blue solution (Figure 11b–d). All spectra have peaks at wavelength of 291 nm, 611 nm, and 656 nm. The intensity of the peaks gradually decreased with time because some factors such as UV, $TiO_2$ nanofibers, and $TiO_2/ZnO$ nanofibers, cause methylene blue to decompose. Since methylene blue shows a low absorbance when photolyzed by photocatalytic reaction, it is possible to know whether the photocatalytic reaction has occurred or not [50,51].

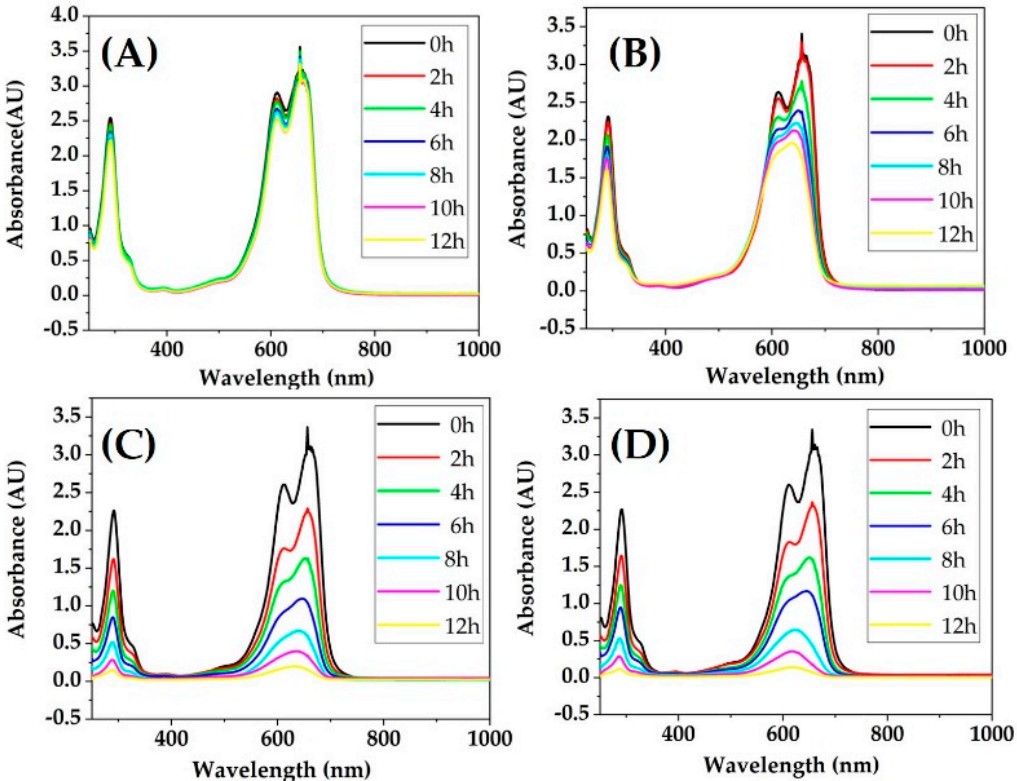

**Figure 11.** Ultraviolet–visible spectroscopy (UV–Vis) spectral change of photocatalytic decomposition of methylene blue over time using diameter-controlled nanofibers: (**A**) UV without nanofibers, (**B**) UV with 0.5 g $TiO_2$ nanofiber added, (**C**) UV with 0.5 g $TiO_2/ZnO$-1 nanofiber added, (**D**) UV with 0.5 g $TiO_2/ZnO$-2 nanofiber added.

To make sure methylene blue decomposed by photocatalytic reaction, the concentration of methylene blue on time was measured and normalized. The curves of normalized value (C/Co) were plotted as shown in Figure 12. In methylene blue solution without $TiO_2$ nanofibers, the concentration very slowly decreased with time and only UV irradiation decomposed the methylene blue. After UV irradiation of 12 h, the normalized value of 0.874 was shown and the photodegradation rate was 12.6%. In methylene blue solution with $TiO_2$ nanofibers, the normalized value decreased to 0.581 and the photodegradation rate was 41.9%. In methylene blue solution with $TiO_2/ZnO$ nanofibers, the normalized value dramatically decreased. This reveals that UV irradiation and photocatalytic reaction of $TiO_2/ZnO$ nanofibers affect the decomposition of methylene blue. After UV irradiation of 12 h, the normalized values in methylene blue with $TiO_2/ZnO$-1 nanofibers and $TiO_2$-ZnO-2 nanofibers were 0.054 and 0.037 and a photodegradation rate of 94.6% and 96.3% were observed, respectively. The photodegradation rate in methylene blue with $TiO_2/ZnO$ nanofibers was higher than with $TiO_2$ nanofibers. This shows that $TiO_2/ZnO$ nanofibers decompose methylene blue more efficiently than $TiO_2$ nanofibers, and the photodegradability of $TiO_2/ZnO$ nanofibers is better than that of $TiO_2$ nanofibers. The best photocatalytic decomposition performance for methylene blue was observed in methylene blue with $TiO_2/ZnO$-2 nanofibers. It was found that $TiO_2/ZnO$-2 nanofibers with a small diameter have better photolytic properties than $TiO_2/ZnO$-1 nanofibers [46–51].

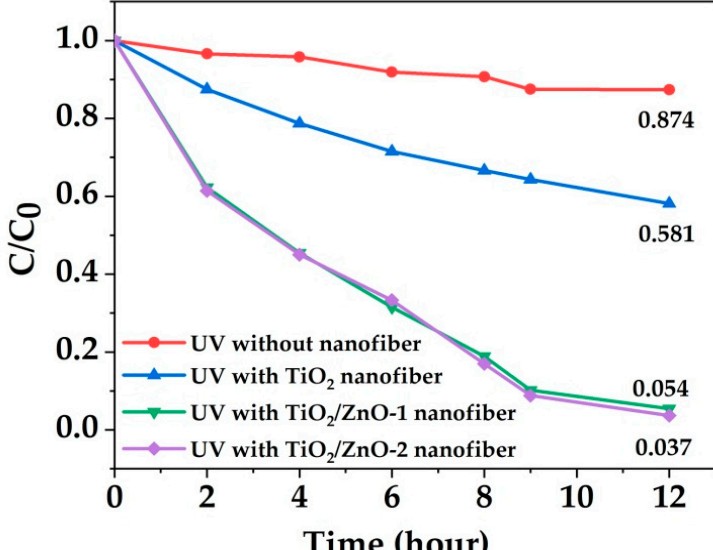

**Figure 12.** Photodegradation efficiency of TiO$_2$/ZnO nanofibers under UV irradiation for the decomposition of methylene blue.

## 4. Conclusions

One-dimensional TiO$_2$/ZnO nanofibers were fabricated by the electrospinning process with various process conditions. The voltage, inflow rate, and amount of acetic acid in precursor were controlled to obtain the optimized microstructure of TiO$_2$/ZnO nanofibers. The average diameter of TiO$_2$/ZnO nanofibers decreased with an increase in the applied voltage and amount of acetic acid and increased with inflow rate. Electrospun TiO$_2$/ZnO nanofibers have the crystalline structures of TiO$_2$ anatase, ZnO, Ti$_2$O$_3$, and ZnTiO$_3$. The best photocatalytic decomposition performance of methylene blue was observed in TiO$_2$/ZnO-2 nanofibers having a diameter of 400 nm. To increase the specific surface area in a way that improves the photocatalytic properties of these TiO$_2$/ZnO nanofibers, variables of the electrospinning process should be controlled to make the diameter small. Electrospun TiO$_2$/ZnO nanofibers with a small diameter will be excellent in photocatalytic properties and bio-friendly properties generated on a large specific surface area and will be utilized in various fields as well as in air pollution removal filters.

**Author Contributions:** Conceptualization, D.-C.P. and W.-Y.C.; methodology, C.-G.L.; validation, C.-G.L., K.-H.N., and W.-T.K.; formal analysis, C.-G.L.; investigation, C.-G.L.; resources, W.-H.Y. and W.-Y.C.; data curation, C.-G.L.; writing—original draft preparation, C.-G.L.; writing—review and editing, W.-Y.C.; visualization, K.-H.N. and W.-T.K.; supervision, W.-Y.C.; project administration, W.-H.Y. and W.-Y.C.; funding acquisition, D.-C.P. and W.-Y.C.

**Funding:** This work was supported by Gangneung-Wonju National University, Korea Institute of Energy Technology Evaluation and Planning (KETEP) and the Ministry of Trade, Industry and Energy (MOTIE) (Grant No. 20181110200070), Korea Agency for Infrastructure Technology Advancement under Construction Technology R&D project (Grant No. 18SCIP-B146646-01), and National Research Foundation of Korea (Grant No. 2019R1I1A3A01057765).

**Acknowledgments:** The authors thank the researchers (Tae-Hyeob Song and Jisun Park) of the Korea Institute of Civil Engineering and Building Technology, for their time and contributions to the study.

**Conflicts of Interest:** The authors declare no conflicts of interest.

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
