# Peer review of "TiO2/ZnO Nanofibers Prepared by Electrospinning and Their Photocatalytic Degradation of Methylene Blue Compared with TiO2 Nanofibers"

_applsci, doi:10.3390/app9163404_

Round 1
Reviewer 1 Report
The authors reported the fabrication of TiO2/ZnO nanofibers with various process conditions and investigated the photocatalytic properties.
Before discussing the scientific contents, the reviewer points out some contradictions in their manuscript.
Figure 2 is “voltage” dependency and figure 4 is “flow rate” dependency. According to Table1, voltage dependency was investigated at flow rate 1 and acetic acid 21 wt% and flow-rate dependency was at voltage 15 kV and acetic acid 21 wt%. However, the SEM images in Fig. 2(a) and Fig. 4(a) were the same. There seems to be similar contradictions. Which of the description is true?
Because of these problems, it is hard to convince the reviewer.
The reviewer has some questions about nanofibers structures.
Some nanowires showed very short length while the others had long continuous structure, for example, comparison between Fig. 4(C) and Fig. 4(A).
In Fig. 4(C), there is variety of the diameter of nanofibers in SEM image, but, no error bar at 0.6 flow rate in Fig. 5. What is definition of the average diameter?
Author Response
The reply was attached as a PDF file.

Reviewer 2 Report
The authors propose a method for the preparation of composite nanofibers composed by zinc oxide and titanium oxide. The material obtained, once characterized, was subsequently used in the photocatalized reduction of methylene blue.
The article is well written and the composite material obtained is interesting and well characterized.
But, both the technique of preparation and the material obtained are not new, in fact there are several works in this regard. In particular, this 2010 work is almost similar: “R. Liu et al. / Materials Chemistry and Physics121 (2010) 432–439”
In addition to having to be mentioned in the references, it also has an almost identical title. I suggest to the authors, for a better distinction of their work, to change something in the title.
Moreover, I would have added a table with catalyst recycling tests.
In conclusion I recommend the publication of the article after a minor revision and in particular after:
1) Title adjustment
2) Insert a table with the recycling tests
Author Response
The reply was attached as a PDF file.

Reviewer 3 Report
The authors reported electro-spun TiO2/ZnO nanofibers for photocatalytic degradation of methylene blue under UV light irradiation. The results are interesting. However, the materials and photocatalytic mechanisms are not well studied. Further experiments are still required to improve the quality of the manuscript. Thus, I recommend a major revision before considering the possible publication of this manuscript in Applied Science.
1. The discussions on the SEM images (e.g. Figures 2, 4, 6, 9) are too simple. It is suggested to add more discussions to show readers more comprehensive information.
2. It is suggested to use different symbols to label the peaks of different materials in the XRD pattern shown in Figure 8. The current labels are a bit confused.
3. In Page 7, the authors claimed “The peaks of … Ti2O2”, but there is no any “Ti2O2” peaks shown in Figure 8. Similarly, the authors also mentioned “Ti2O2” in the abstract and conclusion. Is it a typo?
4. The molar ratios of TiO2: ZnO of the samples of TiO2/ZnO-1 and TiO2/ZnO-2 should be determined.
5. Figure 11 is not a standard version to show the photocatalytic activity. The authors should develop a standard curve to convert the intensities of absorbance to the amounts of methylene blue in the solution, and then calculate the degradation rates(C/C0).
6. The formation of heterojunction in the TiO2/ZnO nanofibers should be important to promote charge separation, which may be the key reason to improve the photocatalytic activity. Thus, the positions of the conduction bands and valance bands of TiO2 and ZnO should be determined, and the formation of heterojunction may be proposed to understand the mechanism. The authors may refer to a heterojunction review paper for more information (J. Mater. Sci. Technol. 2017, 33, 1-22).
Author Response
The reply was attached as a PDF file.

Round 2
Reviewer 1 Report
The paper is now ready for publication. There are a number of typos which I believe a professional editing service would have caught. After revision, this paper can be published.
Reviewer 3 Report
The authors have addressed all comments from reviewers and the quality of the manuscript has been improved. Thus, the current version can be accepted for publication.